# *Rickettsiales* in Italy

**DOI:** 10.3390/pathogens10020181

**Published:** 2021-02-08

**Authors:** Cristoforo Guccione, Claudia Colomba, Manlio Tolomeo, Marcello Trizzino, Chiara Iaria, Antonio Cascio

**Affiliations:** 1Department of Health Promotion, Mother and Child Care, Internal Medicine and Medical Specialties- University of Palermo, 90127 Palermo, Italy; cristoforo.guccione@you.unipa.it (C.G.); claudia.colomba@unipa.it (C.C.); mtolomeo@hotmail.com (M.T.); 2Infectious and Tropical Disease Unit, AOU Policlinico “P. Giaccone”, 90127 Palermo, Italy; marctrix@gmail.com; 3Infectious Diseases Unit, ARNAS Civico-Di Cristina-Benfratelli Hospital, 90127 Palermo, Italy; iaria.chiara@gmail.com

**Keywords:** *Rickettsiales*, *Rickettsia*, *Ehrlichia*, *Orientia*, *Anaplasma*

## Abstract

There is no updated information on the spread of *Rickettsiales* in Italy. The purpose of our study is to take stock of the situation on *Rickettsiales* in Italy by focusing attention on the species identified by molecular methods in humans, in bloodsucking arthropods that could potentially attack humans, and in animals, possible hosts of these *Rickettsiales*. A computerized search without language restriction was conducted using PubMed updated as of December 31, 2020. The Preferred Reporting Items for Systematic Reviews and Meta-Analyses (PRISMA) methodology was followed. Overall, 36 species of microorganisms belonging to *Rickettsiales* were found. The only species identified in human tissues were *Anaplasma phagocytophilum,*
*Rickettsia conorii, R. conorii subsp. israelensis, R. monacensis, R. massiliae,* and *R. slovaca.* Microorganisms transmissible by bloodsucking arthropods could cause humans pathologies not yet well characterized. It should become routine to study the pathogens present in ticks that have bitten a man and at the same time that molecular studies for the search for *Rickettsiales* can be performed routinely in people who have suffered bites from bloodsucking arthropods.

## 1. Introduction

*Rickettsiales* is an order of α-proteobacteria characterized by intracellular tropism with a wide variety of hosts. They are small, gram-negative bacteria that reside free in the host cell cytoplasm, and some of them can be transmitted to human hosts by arthropod vectors such as ticks, lice, fleas, and mites. As suggested by Szokoli et al. we considered included in this order only 3 families: Rickettsiaceae, Anaplasmataceae, and Candidatus Midichloriaceae. *Rickettsiales* encompass human and animal pathogens as well a lot of endosymbiont of arthropods, helminths, and algae with various, pathogenic or not manifestation in the host. The family Rickettsiaceae includes 2 genera: *Rickettsia* and *Orientia.* A modern classification based on whole-genome analysis divides the species of the genus *Rickettsia* in four groups: spotted fever group (*R. rickettsii, R. conorii, R. parkeri*, and several others), typhus group (*R. prowazekii* and *R. typhi*), ancestral group (*R. bellii* and *R. canadensis*, not known to be pathogenic), and transitional group (*R. akari, R. australis*, and *R. felis*) [1,2,3] *Orientia tsutsugamushi* is the etiologic agent of scrub typhus, a rickettsiosis that is widespread in Asia, the islands of the western Pacific and Indian Oceans, and foci in northern Australia [4]. The family Anaplasmataceae includes the genera *Ehrlichia, Anaplasma, Wolbachia*, and *Neorickettsia*. Only the members of the first two genera have been associated to human diseases. The genus *Ehrlichia* includes six species: *E. canis, E. chaffeensis, E. ewingii, E. muris, E. ovis, and E. ruminantium*. The genus *Anaplasma* includes *A. marginale, A. centrale, A. ovis, A. mesaeterum, A. platys*, and *A. phagocytophilum*; only the last is associated to human diseases. The family Candidatus Midichloriaceae does not include any bacteria associated to human disease.

Almost all the cases of human rickettsial diseases in Italy are cases of Mediterranean spotted fever (MSF) caused by *R. conorii* transmitted by the brown dog tick *Rhipicephalus sanguineus.* In Italy, about 400 cases of MSF are reported every year, most of which in people residing in Sicily, Sardinia and Southern Italy with a lethality of less than 3% [5] However, other pathologies such as Tibola/Debonel (Tick Borne Lymphadenopathy/Dermacentor Borne Necrosis Erythema and Lymphadenopathy [6]) and many other *Rickettsia* spp. or subspecies have been identified in recent years in humans, vector arthropods and animals [6]. Other rickettsioses that have been historically documented in Italy are murine typhus and epidemic typhus [5]. Since 1950, only sporadic cases of murine typhus have been reported, and Italy currently appears to be free of epidemic typhus. As in other European countries, imported cases of rickettsial pox, African tick-bite fever (ATBF), and scrub typhus have been reported [5]. 

The purpose of this study is to take stock of the situation on *Rickettsiales* in Italy by focusing attention on the genera until now identified by molecular methods in humans, in bloodsucking arthropods that could potentially attack humans, and in animals possible hosts of these *Rickettsiales*. Our research has therefore mainly focused on the genera *Rickettsia*, *Anaplasma*, *Ehrlichia,* and *Orientia* as these are the ones notoriously associated with human pathology until now.

## 2. Materials and Methods

For the writing of this review a computerized search without language restriction was conducted using PubMed. The search was performed combining the terms “Ricketts * AND Italy”, “Ehrlichi * AND Italy” and “Anaplasma AND Italy”, Orientia AND Italy”. The Preferred Reporting Items for Systematic Reviews and Meta-Analyses (PRISMA) methodology was followed [7]. Only studies that provided data about *Rickettsiales* identified by molecular methods in Italy were included in the review. All molecular methods which reached the species level were considered. A flow chart summarizing the literature research approach is reported in Figure 1.

## 3. Results

A total of 818 papers were retrieved by our search, of these 220 were duplicate and removed; the remains were assessed through their title and abstract and so other 273 were excluded; the selected 325 articles were assessed for eligibility through full text analysis and 168 were excluded as reported in Figure 1; finally, 157 published from 1997 to 2021 studies were included in this review.

The results of our search could be divided in four sections and are analytically reported in Table 1, Table 2, Table 3 and Table 4.

A total of 36 different *Rickettsiales* species belonging to genus *Anaplasma, Ehrlichia* and *Rickettsia;* never *Orientia* spp. were reported in Italy. 32 of them were identified in arthropods, 9 in animal samples, and 10 in human samples (Table 1, Table 2, Table 3 and Table 4).

### 3.1. Rickettsiales and Arthropod Vectors

*Rickettsiales* were identified in 29 species of arthropods, most of them were *Ixodidae* ticks, and 4 species of fleas. The reports present in the scientific literature are resumed in Table 1 and Table 2. Table 1 offer a view centered on the microorganism, for each *Rickettsiales* we report the known association with arthropods and from where it was collected. Indeed, Table 2 offers a point of view centered on the arthropods and for each we report which microorganism and host were associated.

### 3.2. Rickettsiales Identified in Animals

*Rickettsiales* have been identified 179 times in various animal infections, most of which were *Anaplasma* spp. especially in livestock, and *R. conorii* and *E. canis* especially in companion animals. Fifteen species of mammals with or without symptoms were found infected with *Rickettsiales* most of them where *A. phagocytophilum* and *A. platy.* Symptomatic animals were most often pets, with fever and blood count abnormalities (CBC) being the most frequently observed clinical findings; while asymptomatic animals were more often livestock. In Table 3 are resumed the findings in animal samples with clinical manifestations and the number of animals found positive for each *Rickettsiales.* When the original study was done on asymptomatic animals, with the aim of screening, we report also the number of total tested animals and the percentage of prevalence; when the studies was more than one, we report the highest and lowest percentage.

### 3.3. Rickettsiales Involved in Human Disease

*Rickettsiales* were detected 29 times in samples from human patients: 6 cases of anaplasmosis, and 23 cases of rickettsiosis. *Rickettsiales* species identified from human sample and their clinical manifestation are resumed in Table 4. *Rickettsia* spp. associated with MSF were *R. conorii, R. conorii subsp. israelensis, R. conorii subsp. indica, R. massiliae, R. slovaca, and R. monacensis*. *Rickettsia* spp. associated with TIBOLA/DEBONEL were *R. slovaca,* and *R. massiliae*. *R. africae* was identified only once in a traveler from Zimbabwe. *R. aeschlimannii* was associated to a case of acute hepatitis. *A. phagocytophilum* was identified in 6 cases of human illness.

Symptoms mostly associated with MSF were fever, maculopapular rash, and the presence of a necrotic eschar in site of the tick bite “*tache noire*” in French black spot. Cases of MSF caused by *R. conorii subsp. israelensis* were more severe, the rash was petechial and the *tache noire* was not always present. TIBOLA was characterized by the presence of an eschar in the scalp, and enlargement of suboccipital or neck lymph nodes; the eschar in the scalp typically resulted in an area of alopecia.

All the Italian case reports, with the identification of a *Rickettsiales* with molecular method, until species level are reported in Table 4 with the clinical manifestations and number of cases.

## 4. Discussion

The purpose of this article was to analyze all *Rickettsiales* identified in Italy and which could potentially cause disease in humans and to suggest doctors check whether *Rickettsiales* that infect arthropods or the animals they parasite can cause disease in man.

In this section, the findings of the single *Rickettsiales* species are analytically discussed. 

### 4.1. Anaplasma spp.

*Anaplasma *spp.** identified in Italy were *A. marginale, A. ovis, A. platy* and *A. phagocytophilum, A. centrale,* and *A. bovis*. The latter two were found only in sample from animals [98,99,100]. The other four, with the exception of *A. platy*, found only in ticks, were identified both in ticks and fleas. *A. marginale* and *A. ovis* were not a common detection the first was found in the ticks *Haemaphysalis punctata* and *Rhipicephalus turanicus* [93], and in the flea *Xenopsylla cheopis* [9]; the second in the ticks *Ha. punctata* [10], *Rhipicephalus bursa* [11] and in two fleas *X. cheopis* and *Ctenocephalides canis* [9].

The majority of largest report are about *A. phagocytophilum*, found in fleas as *X. cheopis* [9] and ticks belonging to *Ha. punctata* [10], *Hyalomma marginatum* from migratory birds [11], different species of Rhipicephalus like *Rh. Bursa* [13], *Rh. turanicus* [13,27] and *Rh. Sanguineus* [13,26,96]; while it was very often found in *Ixodes* ticks, of these the most common was *I. ricinus*. *Ixodes* spp. is the most diffused tick genera in Italy, it is present almost in every Italian region and climatic areas, from island to continental Italy and in both Tyrrhenian and Adriatic coast. *Ixodes* spp. was found infected with almost all *Rickettsiales,* from the most to the less common, also with the apparent foreign *R. africae* [36] and *R. felis* [48], usually most common in fleas than ticks. Furthermore, *I. ricinus* is the only tick in which *Candidatus Ehrlichia walkerii* was found in Italy [12,16,23,33]. *I. ricinus* was not found mostly on one animal than another, however, seems that it the only tick studied in Italy to be infected when feeding on lizards [78]. *I. ventalloi* is a tick collected from small animals and found infected both with *Ehrlichia, Anaplasma* and *Rickettsia* in Sicily [35,75] and Tuscany [50]; it has also been found in southern Italy to feed on humans [51]. *I. acuminatus* and *I. festai* are rare and have been found infected with some *Anaplasma* spp. [13,32]. Lastly, *A. platy* is a common detection in animal samples, and it was detected less frequently in arthropods; it was found in *Hy. marginatum* [11] from migratory bird, *I. hexagonus* [12] and in same tick belonging to *Rhipicephalus* [11,29,30,31]. Furthermore, co-infection by *A. phagocytophilum* and *R. monacensis* was detected in *I. ricinus* [22]. No animals are an evident favorite host for *Anaplasma* infected ticks. Studies about animal infection with *Anaplasma* spp. are prevalently screening ones conducted on livestock; however, there were also studies about symptomatic animals; *A. phagocytophilum* was identified in horses with flu-like presentation and in some cases with anemia, thrombocytopenia, jaundice, anorexia and leukocytosis [26,96,118,120,121]; *A. phagocytophilum* and *A. ovis* were identified in sheep with a poor general health condition [104] *A. phagocytophilum* was also identified in cows with acute anaplasmosis and presentation that includes hypo-galactia, mucosal paleness, fever and depression [98]. Other cases were diagnosed in pets, mostly infected by *A. phagocytophilum* and *A. platy* [28,29,102,103,108,112,126] less often infected by *A. ovis,* and *A. marginale* in screening studies in asymptomatic dogs [102]. *A. phagocytophilum* was found both in cats and dogs in which depression, fever, weakness and CBC abnormalities like thrombocytopenia, leukocytosis and neutrophilia were described [26,96,107,108,110,111,143] *A. platy* was found twice in cats [136], but there were no differences in the clinical presentation between cats and dogs; *A. platy* infects platelets and classically causes also thrombocytopenia, and monocytosis or neutropenia [29,96,126,127,128,129,130,131,132,134,135,136]. *A. phagocytophilum* was found both in cats and dogs, but more commonly in cats than *A. platy*; less common are the severe thrombocytopenia, and the symptoms were more non-specific. 

Human granulocytic anaplasmosis (HGA) caused by *A. phagocytophilum* in Europe is not uncommon since the first identification of human illness linked to it in Slovenia in 1997, and human positivity before in 1995. Furthermore, serological surveys show that the illness could be underreported and a good number of asymptomatic patients do not have a diagnosis of anaplasmosis [164]. In humans the most common clinical presentation of anaplasmosis is febrile illness, with fever, weakness and sometimes CBC abnormalities [165] without rash or eschar in the site of tick bite. Differently to rickettsiosis, the clinical course can be subacute and persist for months. In Italy, cases of HGA were diagnosed in northeastern Italy, Sardinia and Sicily; of note the case of a patient misdiagnosed for months and treated also for depression before the correct diagnosis was achieved [163,164,165].

### 4.2. Ehrlichia spp.

In Italy *Ehrlichia* spp. has never been identified in human samples. Worldwide, *Ehrlichia* spp. is more often associated with canine pathology. In United States *E. chaffeensis* is the agent human monocytic ehrlichiosis and *E. ewingii*, a canine pathogen, cause of human illness only in immunodeficient or immunosuppressed patients [166].

In arthropods, three *Ehrlichia* spp. were identified in Italy, once *E. ovina* in a tick collected from a healthy sheep [10], more often *E. canis* both in ticks and fleas and *Candidatus Ehrlichia walkerii,* found in *I. ricinus* only in the northernmost regions [12,16,23,33]. *E. canis* was more commonly found in *Rhipicephalus* [12,27,34,46], *Haemaphysalis* [27,32,33], *Hyalomma* [34] and *Dermacentor* [27,34]; only once in *Ixodes* ticks, namely in *I. ventalloi* collected from a cat [35]. *Haemaphysalis* ticks carry prevalently *Ehrlichia* and *Anaplasma*; the genus is not very common, more often found in South and insular Italy. Three tick genera were found infected: *Ha. punctata, Ha. sulcata* more frequently, and only once *Ha. inermis* [51], nonetheless this latter is the only of the three collected from humans. *Ha. punctata*, the commonest species, seems to prefer the livestock and carry often *Anaplasma* [10,32,93] and *Ehrlichia* [10,32,97]. *Ha. sulcata* was found only in Sardinia and carries only two species: *E. canis* [27,34] and *R. hoogstraalii* [45,46]. Not frequently, *Ehrlichia* was found also in ticks non endemic in Italy, collected from migratory birds like *Amblyomma* spp. [36], *Hyalomma rufipes* [36] and *Hy. marginatum* [36,67]. No animal host preference is evident for ticks infected by *Ehrlichia.*

*E. canis*, identified in Italy only in samples from dogs, is the etiological agent of canine monocytic ehrlichiosis (CME), typically characterized by fever, depression, anorexia lymph adenomegaly, splenomegaly, hemorrhagic tendencies, pale mucosa, weight loss, ophthalmologic lesions, neurologic disorders, CBC abnormalities like anemia, leukopenia with lymphocytosis, hypoalbuminemia with hyperglobulinemia and increase in alanine aminotransferase, alkaline phosphatase and C-reactive protein [111,126,132,133,134,136,139,140,141,142]. Of note, *E. canis* in Venezuela has been identified in blood of humans with clinical signs compatible with human monocytic ehrlichiosis [167,168]. Furthermore, *E. ruminantium,* known as ruminant pathogen; has been recently considered an emergent pathogen for human after the report of three deaths associated with it in Africa [169].

### 4.3. Rickettsia spp.

#### 4.3.1. *R. africae*

It is common in Sub-Saharan Africa and South Africa; in Italy it is a recent finding. Indeed, it was found in ticks endemic of African continent, like *Amblyomma* and *Hyalomma*, more often removed from migratory birds [19,36,39] and less often from terricolous animals like sheep and cattle [38,40]. *R. africae* has been identified also in *I. ricinus* removed from migratory birds in Italy [36]. *Amblyomma* has been recently introduced in Italy. Recent studies have documented that this tick can reproduce and could be became endemic also in Italy [37]. 

The human illness associated to *R. africae* is the African Tick Bite Fever (ATBF), similar to MSF but milder and without maculopapular rash; sometimes the eschars may be two. Occasionally, it can cause neuropathy [170]. *R. africae* has been identified in Italy in a woman returning from Zimbabwe, with fever, *tache noire* and rash in the limb ipsilateral to the eschar; the symptomatology was identified as a sacral syndrome, evident in the same side of the eschar [161].

#### 4.3.2. *R. aeschlimannii*

It is often identified in *Hyalomma* ticks removed in small and big animals [32,34,36,38,41,43,44,45,46,48] and less commonly in *Hyalomma* ticks removed from humans [32,42,51], less common it was identified in other ticks as *Amblyomma* [36,41], *D. marginatus* [42], *I. ricinus* [42,52,53], and *R. turanicus* [53]. Its main host, *Hyalomma*, is an African tick typically found when feeding on migratory birds [36,39,41], nonetheless it is common to find these ticks in terricolous animal like sheep, wild boar, or other. It is usually found in Italy in the Tyrrhenian coast on the route of migratory birds. The species *of Hyalomma* found infected in Italy were *Hy. marginatum, Hy. rufipes, Hy. lusitanicum, Hy. detritum, Hy. sulcata, Hy. truncatum. R. aeschlimannii* was found mainly in Tyrrhenian Italy, on the route of migratory birds. Furthermore, *R. aeschlimannii* was identified in *A. marmoreum* [41], another African tick, removed from migratory birds. The first findings of *R. aeschlimannii* was in a *Hyalomma* tick in Morocco, Zimbabwe, Mali and Niger in 1996 [171,172]. The first report of human infection dates back to 2000 in a French traveler returning from Morocco; clinical findings were fever, *tache noire*, and elevated serum liver enzymes; the only Italian case was reported in a man with a strong increase in hepatic enzymes [162]. In the above case, R. *aeschlimannii* was identified in the liver biopsy. PCR on whole blood was negative, differently to the case reported in France. Of note, *R. aeschlimannii* was also identified in the skin of a Greek patient with a single skin manifestation similar to “erythema chronicum migrans” of Lyme disease [173].

#### 4.3.3. *R. conorii*

It is the *Rickettsia* spp. classically associated with MSF. *R. conorii subsp. israelensis and R. conorii subsp. indica* have also been associated with MSF in Italy. 

*R. conorii* has been identified in domestic and wild animals, in domestic dogs, and in wild in a road killed otter [145]. *R. conorii* in dogs has been associated with illness in dogs, with fever, anemia, and thrombocytopenia being the main symptoms, sometimes associated with lethargy [106,112,132,144].

Generally, the clinical symptoms of MSF caused by *R. conorii* begin 4 to 10 days following the tick bite and the signs of the disease may be fever (95%–100%), flu-like symptoms (78%), sore head and muscle aches (64%), skin rash within 6 to 10 days (87%–96%), and eschar (*tache noire*), blackish ulcero-necrotic area at the site of the tick bite (52%–77%). In most subjects, the rash is maculo-papular and also affects the soles of the feet and palms of the hands. The typical signs of these rickettsioses, with the formation of papules, petechiae and rash, are a direct consequence of the colonization and damage of the vascular endothelium by these pathogens. MSF may be complicated by cardiac symptoms (coronary artery ectasia, myocarditis and atrial fibrillation), ocular symptoms (uveitis, retinal vasculitis and retinopathy), neurological symptoms (cerebral infarction, meningoencephalitis have been reported and, sensorineural hearing loss), pancreatic involvement, splenic rupture and acute renal failure, and by hemophagocytic syndrome [146,148,149,150,151,152,153,174,175,176,177,178,179,180,181,182]

MSF caused by *R. conorii subsp. israelensis* is a more severe disease than *R. conorii*’s one; the rash is often petechial and the *tache noire* is almost always absent. Many complications have been reported like neurological involvement [155]. *R. conorii subsp. indica* was identified only once from an inoculation eschar sample of MSF patient in Sicily [154].

#### 4.3.4. *R. helvetica*

*R. helvetica* has been identified mountainous territory, more often in northern Italy and in areas far from the coast. It was identified in *I. ricinus* removed from small animals [22,55,62,67,72,74,76], deers [62,76], vegetation [64,65,66,68,70,71,73] and human [14,23,51,77]. It was also identified in *I. festai* [32,34,45] *I. acuminatus* [50], *I. ventalloi* [50,51,75] and *I. trianguliceps* [62]. However, the geographical distribution of these last three Ixodes is different: *I. acuminatus* was found in central-north Italy, far from the coast. *I. festai* only in Sardinia and *I. trianguliceps* in the eastern alps. Only once it was found in *R. sanguineus* collected in vegetation [68]

In humans, *R. helvetica* infection presents as a mild disease associated with fever, headache, and myalgia but not with a cutaneous rash. In Italy only one human case of disease caused *R. helvetica* presenting with fever, headache, myalgia and arthralgia was diagnosed only by serology [183]. However, *R. helvetica* has been identified in Sweden in two case of meningitis, in one of these *R. helvetica* was identified in the cerebrospinal fluid [182,184]. 

#### 4.3.5. *R. massiliae*

*R. massiliae* belongs to the spotted fever group rickettsiae, and is distributed worldwide. The ticks in which *R. massiliae* was more commonly identified in Italy were *R. sanguineus* [32,42,43,45,46,47,48,49,56,67,79], and *R. turanicus* [42,43,47,48,53]. Less commonly it was found in *I. ricinus* [48,52,53], never this happened in Sicily or in Sardinia. 

The first human case of *R. massiliae* infection was diagnosed in a Sicilian patient with MSF; the second case was in a patient in southern France who had MSF complicated by acute loss of vision; and the third case was in a woman in Argentina who had fever, a palpable purpuric rash, and *tache noire*. Two cases of TIBOLA/DEBONEL caused by *R. massiliae* have been described in Italy: one in in north Italy, the other in Sicily [157,158].

#### 4.3.6. *R. monacensis*

In contrast with the other species, most common in the south and insular regions, *R. monacensis* is most common in the inland. *I. ricinus* [22,48,49,52,54,55,64,66,68,69,70,71,72,74,75,76,78,80,81,82,84] and *I. ventalloi* [75] were found infect by this species. Less often *R. monacensis* was found in *D. marginatus* [51,53], *Ha. punctata* [48,80], *R. sanguineus* [48,68,75], and *R. turanicus* [53]. Furthermore, *R. monacensis* was detected inside *Crataerina pallida*, a hematophagous diptera [54]. Sometimes *R. monacensis* has been found coinfecting a tick, and another time with *R. tamurae* [62]. In humans, *R. monacensis* may cause MSF-like illness as described by Jado et al. [185] in Spain. In Italy, it has been identified only once, in the eschar biopsy of an anaeructive MSF in Sardinia [160].

#### 4.3.7. *R. slovaca* and Other Agents of TIBOLA/DEBONEL/SENLAT

The first identification of *R. slovaca*, and the related illness, was in France in 1996 from a woman bitten by a *D. marginatus* in the scalp; the woman complained of fatigue, lymphadenopathy, fever, eschar with erythematous halo and no rash; later, also *R. raoultii* and *R. rioja* were associated with this syndrome [186,187,188]. 

*Dermacentor* ticks infected with the above rickettsiae were found prevalently in Tyrrhenian coast and western alps and have a period of activity cold season (from late fall to mid spring [186]). The tick species found infected were *D. reticulatus* [89] and *D. marginatus* [32,34,40,42,43,45,46,51,61,72,80,83,84,85,86,87,88,90,92], the first was found only once in the western alps on a wild boar, the second was more commonly found. Other ticks involved in ecology of *R. slovaca* are *Ha. punctata* [40], *Hy. sulcata* [34], *I. ricinus* [48,80,87], *I. hexagonus* [48], and *R. sanguineus* [42,51]. Wild boar appears to be the favorite host for infected *Dermacentor* spp., nonetheless *R. slovaca* was not found in ticks collected more often from one animal than others.

*R. raoultii* [46,51,61,72,83,84,85,86,87] and *R. rioja* [88], were both found in Italy in *D. marginatus* and, *R. raoultii* in other ticks like *Ixodes* spp. [48,52,62], *Rhipicephalus* spp. [46] and *Hyalomma* spp. [41]. 

For TIBOLA and DEBONEL, was proposed by the Marseille group the name SENLAT (scalp eschar and neck lymphadenopathy after tick bite) to bring together the clinical manifestation without etiological differentiation. Indeed, others tick-borne pathogen than *R. slovaca* as *R. massiliae,*
*Bartonella henselae* and *Borrelia burgdorferi* have been associated with this syndrome [157,158,189,190]. 

*R. slovaca* has been documented as agent of TIBOLA also in Italy [92]. Of note, *R. slovaca* was also identified in Sicily in a “MSF like” case [154].

#### 4.3.8. *R. felis*

*R. felis* is typically found in *Ctenocephalides felis* [9,35,58,59,60], the common flea of the cat. Of note, *C. felis* can parasite also other mammals like dogs or foxes. *R. felis* in Italy has been sometimes identified in *I. hexagonus* [48] and in *R. turanicus* [61], but never in humans. The disease caused by *R. felis* is similar to murine typhus, with fever, myalgia, headache, and rash [189,190]; the eschar may be present. Severe complication, like meningoencephalitis, may occur [191]. *R. felis* has also been identified in a cutaneous swab of a Senegalese 8-month-old girl with “yaaf”, a febrile illness associated with a cutaneous eruption [192].

The only Italian case of *R. felis* infection occurred in a traveler from Nepal and was confirmed with indirect fluorescent antibody tests in 2015. The patient complained headache, fever, nausea and vomiting, a raising in liver enzymes was also observed. Interestingly Nepal’s altitude is not well suitable for ticks or fleas, the patients report multiple attack by aquatic leeches, removed with water and salt [193]. The most recent review worldwide that describe the diffusion of *R. felis* was published in 2016 [194]. In consideration of the spread of flea infection found in Italy, it is possible that the disease may be present in Italy even if it is generally not sought.

#### 4.3.9. Other *Rickettsia *spp.**

Other *Rickettsia *spp.** identified in arthropods in Italy were *R. belli* [54] *R. hoogstraalii* [32,45,78], *R. limoniae* [70], *R. peacockii* [51], *R. rhipicephali* [79], *R. sp. Strain S* [40], R. sp. strainTwKm01 [53], *Candidatus R. barbariae* [32,43,47], *Candidatus R. siciliensis* [89], *Candidatus R. mendelii* [74], *R. honei* [40], *R. tamurae* [42], *R. rioja* [88], *R. limoniae* [70], *R. raoultii* [41,46,48,51,52,61,62,72,83,84,85,86,87]. *R. belli*, interestingly, was identified in *Crataerina pallida*, an Hippoboscidae hematophagous dipter [54]. None of the above *Rickettsia* spp. has ever been associated with human disease all over the world.

*R. sibirica mongolotimoniae* and *R. akari* have never been identified in Italy. *R. sibirica mongolotimoniae* is etiological agent of Lymphangitis Associated Rickettsiosis (LAR) [195]. It is frequently associated with *Hyalomma* spp., ticks widely distributed across the Tyrrhenian coast of Italy. Since the discovery in 1996 of a case of human illness associated with it, it has been documented in France, Spain and Greece and other country. The disease could be present also in Italy and for this reason it is under surveillance according to the report of European Centre for Disease Prevention [196].

*R. akari* is the agent of rickettsial pox and is transmitted by the *Lyponyssoides sanguineus,* the house-mouse-mite. Cases of rickettsial pox have been reported from all continents. *R. akari* infection presents with a triad of fever, vesicular rash, and eschar. Between the first and fourth day of fever a papulovesicular eruption occurs on many parts of the body except the palms of the hands and soles of the feet. The eruption is nonpruritic and resolves without leaving scars. In Italy, *R. akari* has never been identified in humans, in mite or in animal [186]. 

*R. prowazekii*, the agent of louse-borne typhus. This disease occurs in colder regions of central and eastern Africa, central and South America, and Asia. In recent years, most outbreaks have taken place in Burundi, Ethiopia and Rwanda. Typhus fever occurs in conditions of overcrowding and poor hygiene, such as in prisons and refugee camps. Cases of louse-borne typhus in Italy were reported before World War II. *R. prowazekii* has never been identified in Italy by molecular methods. Symptoms of epidemic typhus begin within 2 weeks after contact with infected body lice. Signs and symptoms may include: headache, confusion, fever and chills, rapid breathing, cough, vomiting, muscle aches, and rash. *R. prowazekii* can remain dormant for years or even decades in patients who recover from the primary infection. In certain individuals, stress or waning immunity are likely to reactivate this persistent infection, and cause a recrudescent form of typhus known as Brill-Zinsser disease [197]. A case of seroconversion to *R. prowazekii* in a homeless person has been reported in France in 2005 [198]. The current migratory flows from Africa to Italy require us to pay attention to this disease which could reactivate in people exhausted by the travel and the discomfort suffered in the prison camps.

*R. typhi*, the agent of flea-borne typhus. It occurs in tropical and subtropical climates around the world including areas of the United States. Symptoms of flea-borne typhus begin within 2 weeks after contact with infected fleas. Signs and symptoms may include: Fever and chills, body aches and muscle pain, vomiting, cough, and rash that typical occurs around day 5 of illness. Since 1950, only sporadic cases of murine typhus have been reported, and *R. typhi* has never been identified in Italy by molecular methods. However, a case murine typhus diagnosed only by serology in a 75-year-old woman presenting with spotted fever followed by acute renal failure and septic shock was recently described in south Italy [199]. 

### 4.4. Orientia spp.

*Orientia tsutsugamushi* is the etiologic agent of scrub typhus, a rickettsiosis that is widespread in Asia, the islands of the western Pacific and Indian Oceans, and foci in northern Australia. It is transmitted by the bites of larval trombiculid mites (chiggers) of the genus *Leptotrombidium*. Recent evidences from Africa, France, the Middle East, and South America, have led to the supposition that scrub typhus should no longer be considered restricted to Asia and Western Pacific [200]. Besides, cases of travel-associated scrub typhus have been reported from Europe, North America, and Japan [201]. Symptoms of scrub typhus usually begin within 10 days of being bitten. Signs and symptoms generally include: headache, fever and chills, muscle pain, a black eschar in the site of the chigger bite, enlarged lymph nodes and maculopapular rash [202]. In Italy, *Orientia *spp.** has never been identified neither in man nor in animals nor in mites. 

## 5. Conclusions 

Rickettsiales found in humans in Italy were: R. aeschlimannii, R. africae, R. massiliae, R. monacensis, R. slovaca, R. conorii, R. conorii subsp. israelensis, R. conorii subsp. indica and A. phagocytophilum. MSF and TIBOLA and HGA were the most frequent clinical manifestations. E. canis, A. platy and A. phagocytophilum were the most frequently identified Rickettsiales found in dogs and cattle, respectively. Other Rickettsiales identified were: A. bovis, A. ovis, A. marginale, A. centrale, A. platy, E. ovina, Candidatus N. mikurensis, Candidatus R. siciliensis, Candidatus R. barbariae, Candidatus. R. mendelii, R. hoogstraalii, R. limoniae, R. peacockii, R. rhipicephali, R. sp. Strain S, R. sp. strainTwKm01, R. belli, R. tamurae, R. rioja, R. limoniae, R. raoultii, R. honei; some of them, even if it has not yet been demonstrated, could in the future be shown to be capable of causing in humans not yet well characterized syndromic pictures. That’s why molecular studies for the search for Rickettsiales should be routinely performed in people who have been bitten by bloodsucking arthropods. 

## Figures and Tables

**Figure 1 pathogens-10-00181-f001:**
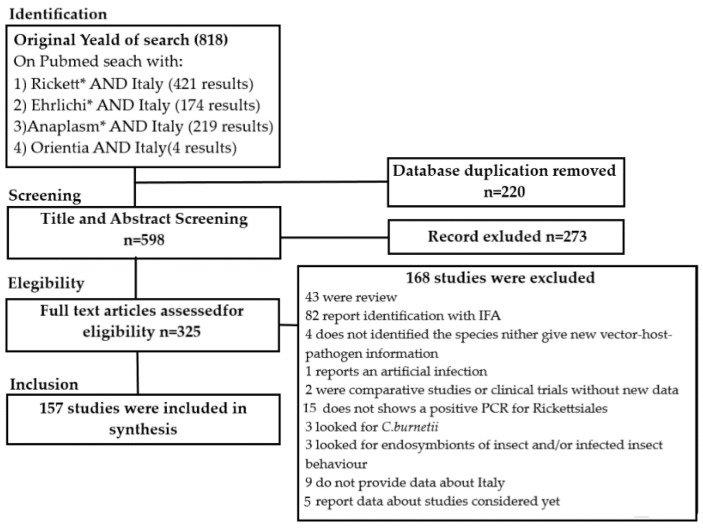
Process of selection of articles for the review according to PRISMA methodology [7]. *: Bibliography.

**Table 1 pathogens-10-00181-t001:** Species of *Rickettsiales*, arthropods from which they were identified and corresponding hosts.

***Rickettsiales***	**Arthropod**	**Collected From**	**Reference**
*Anaplasma marginale*	*Haemaphysalis punctata*	Cattle	[8]
	*Rhipicephalus turanicus*	Cattle	[8]
	*Xenopsylla cheopis*	Fox	[8]
*A. ovis*	*Ctenocephalides canis*	Fox	[9]
	*Haemaphysalis punctata*	Sheep	[10]
	*Rhipicephalus bursa*	Goat, mouflon	[11]
	*Rhipicephalus sanguineus s.l.*	Dog	[12]
	*Xenopsylla cheopis*	Fox	[9]
*A. phagocytophilum*	*Haemaphysalis punctata*	Sheep	[10]
	*Ixodes ricinus*	Bird, cat, dog, fallow deer, free life, horse, human, red deer, rodent, roe deer, sheep, vegetation	[12,13,14,15,16,17,18,19,20,21,22,23,24]
	*Ixodes ventalloi*	Vegetation	[25]
	*Hyalomma marginatum*	Migratory birds	[11]
	*Ixodes acuminatus*	Dog	[13]
	*Rhipicephalus sanguineus*	Dog	[13,26]
	*Rhipicephalus turanicus*	Dog, horse, sheep, goat	[13,27]
	*Rhipicephalus bursa*	Dog	[13]
	*Xenopsylla cheopis*	Fox	[9]
*A. platy*	*Ixodes hexagonus*	Dog	[12,28]
	*Hyalomma marginatum*	Wild boar	[11]
	*Rhipicephalus bursa*	Goat	[11]
	*Rhipicephalus sanguineus*	Dog	[29,30]
	*Rhipicephalus *sp.* II*	Dog	[31]
*Anaplasma *spp.**	*Haemaphysalis punctata*	Mouflon	[32]
	*Ixodes festai*	Hedgehog	[32]
	*Rhipicephalus bursa*	Mouflon, cattle, fox	[32]
	*Rhipicephalus sanguineus s.l.*	Cat, fox, goat, marten, mouflon	[32]
*Candidatus Ehrlichia walkerii*	*Ixodes ricinus*	Dog, goat, human, sheep, vegetation,	[12,16,23,33]
*E. ovina*	*Haemaphysalis punctata*	Sheep	[10]
*E. canis*	*Cediopsylla inaequalis*	Fox	[9]
	*Dermacentor marginatus*	Mouflon, wild boar	[27,34]
	*Haemaphysalis punctata*	Mouflon	[32]
	*Haemaphysalis sulcata*	Goat, mouflon	[27,34]
	*Hyalomma marginatum*	Sheep, swine	[34,35]
	*Ixodes ventalloi*	Cat	[35]
	*Rhipicephalus bursa*	Deer, sheep, goat	[32,34]
	*Rhipicephalus sanguineus s.l.*	Dog, fox, sheep	[12,32,34]
	*Rhipicephalus sanguineus*	Dog	[27]
	*Xenopsylla cheopis*	Fox	[9]
*Ehrlichia *spp.**	*Amblyomma *spp.**	Migratory birds	[36]
	*Hyalomma marginatum*	Migratory birds	[36]
	*Hyalomma rufipes*	Migratory birds	[36]
*Rickettsia africae*	*Amblyomma marginatus*	Sheep	[37]
	*Amblyomma variegatum*	Migratory birds	[38,39]
	*Hyalomma marginatum*	Cattle	[40]
	*Hyalomma rufipes*	Migratory birds	[41]
	*Hyalomma *spp.**	Migratory birds	[36]
	*Ixodes ricinus*	Migratory birds	[36]
*R. aeschlimannii*	*Amblyomma marmoreum*	Migratory birds	[41]
	*Dermacentor marginatus*	Human	[41]
	*Hyalomma marginatum*	Bird, cattle, dog, free life, goat, hedgehog, migratory birds, horse, human, mouflon, red deer, sheep, vegetation	[32,34,36,38,40,42,43,44,45,46,47,48,49,50]
	*Hyalomma lusitanicum*	Human, free life	[42,46,51]
	*Hyalomma rufipes*	Migratory birds	[36,38,39,41]
	*Hyalomma truncatum*	Migratory birds	[38]
	*Hyalomma detritum*	Vegetation	[50]
	*Ixodes ricinus*	Free life, human	[42,52,53]
	*Rhipicephalus turanicus*	Free life	[53]
*R. barbarie (candidatus)*	*Rhipicephalus sanguineus s.l.*	Dog, fox, goat	[32]
	*Rhipicephalus turanicus*	Goat, sheep	[43,47]
*R. belli*	*Hippoboscidae Crataerina pallida*	Bird	[54]
*R. conorii*	*Ixodes ricinus*	Free life	[53]
	*Rhipicephalus turanicus*	Brown Hare, cattle, free life, human	[40,42,48,53]
	*Rhipicephalus sanguineus*	Dog, free life	[49,55,56]
	*Rhipicephalus sanguineus s.l.*	Human	[42]
*R. conorii *subsp.* israelensis*	*Rhipicephalus sanguineus*	Dog	[43,57]
	*Rhipicephalus sanguineus s.l.*	Dog, fox, goat	[32]
*R. felis*	*Ctenocephalides felis*	Dog, cat, fox	[9,35,58,59,60]
	*Ixodes hexagonus*	Fox, hedgehog	[48]
	*Rhipicephalus turanicus*	Sheep	[61]
*R. helvetica*	*Ixodes ricinus*	Bird, cat, dog, fox, free life, human, lizard, migratory birds, red deer, rodent, roe deer, vegetation	[22,23,48,51,52,55,61,62,63,64,65,66,67,68,69,70,71,72,73,74,75,76,77,78]
	*Ixodes ventalloi*	Bird, cat, human	[50,51,75]
	*Ixodes festai*	Cat, hedgehog	[45,46,67]
	*Ixodes acuminatus*	Cat, red partridge	[50]
	*Ixodes trianguliceps*	Rodent	[62]
	*Rhipicephalus sanguineus*	Vegetation	[68]
*R. honei*	*Hyalomma marginatum*	Cattle	[40]
*R. hoogstraalii*	*Haemaphysalis punctata*	Mouflon, sheep	[45,46]
	*Haemaphysalis sulcata*	Mouflon, sheep	[45,46]
	*Ixodes ricinus*	Lizard	[78]
*R. IRS3*	*Ixodes ricinus*	Migratory birds, vegetation	[64,74]
	*Ixodes ventalloi*	Cat	[50]
*R. limoniae*	*Ixodes ricinus*	Vegetation	[70]
*R. massiliae*	*Ixodes ricinus*	Free life, human	[48,52,53]
	*Rhipicephalus turanicus*	Brown hare, cattle, free life, goat, human	[42,43,47,48,53]
	*Rhipicephalus sanguineus s.l.*	Dog, fox, goat, human, sheep	[32,36,42,46,48]
	*Rhipicephalus sanguineus*	Dog, fox, human, cat	[43,45,49,56,79]
*R. mendelii (candidatus)*	*Ixodes ricinus*	Migratory birds	[74]
*R. monacensis*	*Dermacentor marginatus*	Free life, human	[51,53]
	*Haemaphysalis punctata*	Chamois, fallow deer	[48,80]
	*Hippoboscidae Crataerina pallida*	Bird	[54]
	*Ixodes ricinus*	Bear, cat, chamois, dog, fallow deer, free life, goat, hare, human, lizard, migratory birds, red deer, rodent, roe deer, vegetation, wild boar, wolf	[22,48,51,52,53,55,62,64,66,68,69,70,71,72,74,75,76,77,78,80,81,82]
	*Rhipicephalus sanguineus s.l.*	Cat, dog	[48,75]
	*Rhipicephalus turanicus*	Free life	[53]
	*Rhipicephalus sanguineus*	Vegetation	[68]
*R. tamurae*	*Ixodes ricinus*	Human, red deer	[62]
*R. peacockii*	*Dermacentor marginatus*	Human	[51]
*R. raoultii*	*Dermacentor marginatus*	Deer, human, rodent, wild boar	[45,51,61,72,83,84,85,86,87]
	*Hyalomma* spp.	Migratory birds	[41]
	*Ixodes hexagonus*	Badger	[48]
	*Ixodes ricinus*	Free life, red deer	[52,62]
	*Rhipicephalus sanguineus s.l.*	Dog	[46]
*R. rhipicephali*	*Rhipicephalus sanguineus*	Cat	[79]
*R. rioja*	*Dermacentor marginatus*	Vegetation, wild boar	[88]
*Candidatus R. siciliensis*	*Rhipicephalus turanicus*	Human	[89]
*R. slovaca*	*Dermacentor marginatus*	Cattle, chamois, deer, dog, human, red deer, rodent, roe deer, sheep, swine, vegetation, wild boar	[32,34,39,42,43,45,46,48,51,61,72,80,83,84,85,86,87,88,90]
	*Dermacentor reticulatus*	Wild Boar	[88]
	*Haemaphysalis punctata*	Cattle	[40]
	*Hyalomma sulcata*	Dog, sheep	[34]
	*Ixodes ricinus*	Human, red deer, wild boar	[48,80,87]
	*Rhipicephalus sanguineus s.l.*	Human	[42]
	*Rhipicephalus sanguineus*	Human	[51]
*R. sp strain S*	*Hyalomma marginatum*	Cattle	[42]
*R. sp. strain TwKm01*	*Ixodes ricinus*	Free life	[53]
	*Rhipicephalus turanicus*	Free life	[53]
*Rickettsia *spp.**	*Haemaphysalis inermis*	Human	[51]
	*Rhipicephalus annulatus*	Cattle	[91]
	*Rhipicephalus bursa*	Cattle	[91]
	*Rhipicephalus turanicus*	Fox	[49]

**Table 2 pathogens-10-00181-t002:** Arthropod species found infected with *Rickettsiales* in Italy.

Arthropods	*Rickettsiales*	Arthropods Collected From	Reference
*Amblyomma marginatus*	*R. africae*	Sheep	[37]
*Amblyomma marmoreum*	*R. aeschlimannii*	Migratory birds	[41]
*Amblyomma *spp.**	*Ehrlichia *spp.**	Migratory birds	[36]
	*R. aeschlimannii*	Migratory birds	[36]
*Amblyomma variegatum*	*R. africae*	Migratory birds	[38,39]
*Cediopsylla inaequalis*	*E. canis*	Fox	[9]
*Ctenocephalides canis*	*A. ovis*	Fox	[9]
*Ctenocephalides felis*	*R. felis*	Cat, dog, fox	[9,35,58,59]
*Dermacentor marginatus*	*E. canis*	Mouflon, wild boar	[27,34]
	*R. aeschlimannii*	Human	[42]
	*R. monacensis*	Free life, human	[51,53]
	*R. peacockii*	Human	[51]
	*R. raoultii*	Deer, human, rodent, wild boar	[23,46,51,83,84,85,86,87]
	*R. rioja*	Vegetation, wild boar	[88]
	*R. slovaca*	Cattle, chamois, deer, dog, human, red deer, rodent, roe deer, sheep, swine, vegetation, wild boar	[32,34,40,42,43,46,48,51,61,72,80,83,84,85,86,87,88,90,92]
*Dermacentor reticulatus*	*R. slovaca*	Wild Boar	[88]
*Haemaphysalis inermis*	*Rickettsia* spp.	Human	[51]
*Haemaphysalis punctata*	*A. marginale*	Cattle	[8,93]
	*A. ovis*	Sheep	[10]
	*A. phagocytophilum*	Sheep	[10]
	*Anaplasma *spp.**	Mouflon	[32]
	*E. canis*	Mouflon	[32]
	*E. ovina*	Sheep	[10]
	*R. hoogstraalii*	Mouflon, sheep	[32,45]
	*R. monacensis*	Chamois, fallow deer	[48,80]
	*R. slovaca*	Cattle	[40]
*Hippoboscidae Crataerina pallida*	*R. belli*	Bird	[54]
	*R. monacensis*	Bird	[54]
*Hyalomma detritum*	*R. aeschlimannii*	Vegetation	[50]
*Hyalomma lusitanicum*	*R. aeschlimannii*	Human, free life	[42,46,51]
*Hyalomma marginatum*	*A. platy*	Migratory birds	[11]
	*A. phagocytophilum*	Migratory birds	[11]
	*E. canis*	Sheep, swine	[34]
	*Ehrlichia *spp.**	Migratory birds	[36]
	*R. africae*	Cattle	[40]
	*R. aeschlimannii*	Bird, cattle, dog, free life, goat, hedgehog, horse, human, migratory birds, mouflon, red deer, sheep, vegetation	[32,34,36,38,40,42,43,44,45,46,47,48,50]
	*R. honei*	Cattle	[40]
	*R. sp strain S*	Cattle	[40]
*Hyalomma rufipes*	*Ehrlichia *spp.**	Migratory birds	[36]
	*R. africae*	Migratory birds	[41]
	*R. aeschlimannii*	Migratory birds	[36,38,39,41]
*Hyalomma *spp.**	*R. raoultii*	Migratory birds	[41]
*Hyalomma sulcata*	*R. slovaca*	Dog, sheep	[34]
*Hyalomma truncatum*	*R. aeschlimannii*	Migratory birds	[38]
*Ixodes acuminatus*	*A. phagocytophilum*	Dog	[13]
	*R. helvetica*	Cat, red partridge	[50]
*Ixodes hexagonus*	*A. platy*	Dog	[12]
	*R. felis*	Fox, hedgehog	[48]
	*R. raoultii*	Badger	[48]
	*R. slovaca*	Badger	[48]
*Ixodes festai*	*Anaplasma *spp.**	Hedgehog	[32]
	*R. helvetica*	Cat, hedgehog	[32,34,45]
*Ixodes ricinus*	*A. phagocytophilum*	Bird, cat, dog, fallow deer, free life, horse, human, red deer, sheep, vegetation	[12,13,15,16,17,19,20,21,22,23,24,62,94,95]
	*Candidatus Ehrlichia walkerii*	Dog, goat, human, sheep, vegetation, human	[12,16,23,33]
	*R. africae*	Migratory birds	[36]
	*R. aeschlimannii*	Free life, human	[42,52,53]
	*R. conorii*	Free life	[53]
	*R. helvetica*	Bird, cat, dog, fox, free life, human, lizard, migratory birds, red deer, roe deer, vegetation	[23,51,52,55,61,62,63,64,65,66,67,68,69,70,71,72,73,74,76,78]
	*R. hoogstraalii*	Lizard	[78]
	*R. IRS3*	Migratory birds, vegetation	[64,74]
	*R. limoniae*	Vegetation	[70]
	*R. massiliae*	Free life, human	[48,52,53]
	*Candidatus R. mendelii*	Migratory birds	[74]
	*R. monacensis*	Bear, cat, chamois, dog, fallow deer, free life, goat, hare, human, lizard, migratory birds, red deer, roe deer, rodent, vegetation, wild boar, wolf	[22,48,49,51,52,53,55,64,66,68,69,70,71,72,74,75,76,78,80,81,82,84]
	*R. tamurae*	Human, red deer	[62]
	*R. raoultii*	Free life, red deer	[52,62]
	*R. slovaca*	Human, red deer, wild boar	[48,80,87]
	*R. sp. strain TwKm01*	Free life	[53]
*Ixodes trianguliceps*	*R. helvetica*	Rodent	[62]
*Ixodes ventalloi*	*A. phagocytophilum*	Vegetation	[25]
	*E. canis*	Cat	[35]
	*R. helvetica*	Bird, cat, human	[50,51,75]
	*R. IRS3*	Cat	[50]
	*R. monacensis*	Cat	[75]
*Rhipicephalus annulatus*	*Rickettsia *spp.**	Cattle	[91]
*Rhipicephalus bursa*	*A. phagocytophilum*	Dog	[11]
	*A. platy*	Goat	[11]
	*Anaplasma *spp.**	Mouflon	[11]
	*A. ovis*	Goat, mouflon	[11]
	*Anaplasma *spp.**	Cattle, fox, goat	[11]
	*E. canis*	Deer, goat, sheep	[32,34]
	*Rickettsia *spp.**	Cattle	[91]
*Rhipicephalus sanguineus*	*A. phagocytophilum*	Dog	[13,96]
	*A. platy*	Dog	[29,30]
	*E. canis*	Dog	[27]
	*R. conorii*	Dog, free life	[49,55,56]
	*R. conorii subsp. israelensis*	Dog	[43,57]
	*R. helvetica*	Vegetation	[68]
	*R. massiliae*	Cat, fox, human	[43,45,47,49,56,79]
	*R. rhipicephali*	Cat	[79]
	*R. monacensis*	Vegetation	[68]
	*R. slovaca*	Human	
*Rhipicephalus sanguineus s.l*	*A. ovis*	Dog	[12]
	*Anaplasma* spp.	Cat, fox, goat, marten, mouflon	[32]
	*E. canis*	Dog, fox, sheep	[12,32,34]
	*Candidatus R. barbariae*	Dog, fox, human	[32]
	*R. conorii*	Human	[42]
	*R. conorii subsp. israelensis*	Dog, fox, goat	[32]
	*R. massiliae*	Dog, fox, goat, human, sheep	[32,34,42,46,48]
	*R. monacensis*	Cat, dog	[48,75]
	*R. raoultii*	Dog	[46]
	*R. slovaca*	Human	[42]
*Rhipicephalus sp. II*	*A. platy*	Dog	[31]
*Rhipicephalus turanicus*	*A. marginale*	Cattle	[8]
	*A. phagocytophilum*	Dog, horse, sheep, goat	[13,27]
	*R. aeschlimannii*	Free life	[53]
	*Candidatus R. barbariae*	Goat, sheep	[43]
	*R. conorii*	Brown Hare, cattle, free life, human	[40,42,48,53]
	*R. felis*	Sheep	[61]
	*R. massiliae*	Brown Hare, cattle, free life, goat, human	[42,43,47,48,53]
	*R. monacensis*	Free life	[53]
	*Candidatus R. siciliensis*	Human	[89]
	*R. sp. strain TwKm01*	Free life	[53]
	*Rickettsia* spp.	Fox	[97]
*Xenopsylla cheopis*	*A. marginale*	Fox	[9]
	*A. ovis*	Fox	[9]
	*A. phagocytophilum*	Fox	[9]
	*E. canis*	Fox	[9]

**Table 3 pathogens-10-00181-t003:** *Rickettsiales* identified in wild and domestic animal tissues in Italy.

*Rickettsiales*	Animal	Clinical Manifestation	Number of Positive/Tested Animals (%)	Reference
*Anaplasma bovis*	Cattle	No symptoms	1/51 (1.9%)	[98]
	Sheep	No symptoms	3/20 (15%)	[99]
*A. centrale*	Cattle	Acute anaplasmosis: hypo-galactia, mucosal paleness, depression high temperature (40–45 °C), anemia, thrombocytopenia, erythrocytic inclusion	26/270 (8–21%)	[98,100]
*A. marginale*	Cattle	No symptoms	535/2500 (2.3–76,4%)	[8,93,98,99,100,101,102,103,104,105]
	Dog	No symptoms	2/46 (4.3%)	[102]
	Goat	No symptoms	95/184 (27–85%)	[99,102,106]
	Horse	No symptoms	26/134 (19.4%)	[102]
	Rodent	No symptoms	3/69 (4.3%)	[102]
	Sheep	No symptoms	35/286 (12.2%)	[102]
*A. ovis*	Cattle	No symptoms	22/374 (5.9%)	[102]
	Dog	No symptoms	2/46 (4.3%)	[102]
	Goat	No symptoms	113/468 (14.9–85%)	[99,101,102]
	Horse	No symptoms	23/134 (17.1%)	[102]
	Rodent	No symptoms	23/69 (33%)	[102]
	Sheep	No symptoms	164/716 (11–81.8%)	[10,93,99,101,102,103,104]
*A. phagocytophilum*	Cattle	No symptoms	233/984 (2–88%)	[99,100,101,102,105,106]
	Cat	Lymphadenomegaly, pale mucous, stomatitis, sign of respiratory involvement	47/360 (1–31.9%)	[101,107]
	Chamois	No symptoms	6/9 (66.6%)	[19]
	Dog	Fever, acute lameness to right forelimb, depression, jaundice, dysorexia, leukocytosis, neutrophilia, thrombocytopenia, mild anemia	15	[26,96,102,107,108,109,110,111,112]
	Donkey	No symptoms	3/3 (100%)	[93]
	Fallow deer	No symptoms	42/80 (40%)	[24,113]
	Fox	No symptoms	18/277 (0.8–10.8%)	[114,115,116]
	Goat	No symptoms	55/203 (16.9–72%)	[103,117]
	Horse	Hyperthermia, anemia, anorexia, jaundice, myalgia, reluctance to move, thrombocytopenia, leukocytosis	45	[26,96,103,118,119,120,121,122]
	Red deer	No symptoms	66/119 (54–59%)	[116,123]
	Rodent	No symptoms	56/2259 (3–5.3%)	[12,103,124]
	Roe deer	No symptoms	32/116 (19–75%)	[103,116,125]
	Sheep	Screening or poor health condition	166/1496 (0.1–81%)	[10,99,103,104,117,118,119]
*A. platy*	Dog	Depression, myalgia, anorexia, fever, epistaxis, rough coat, reluctance to move, diarrhea, lymphadenomegaly, weight loss, pale mucous membranes, high hepatic enzymes, splenomegaly, ascites, thrombocytopenia, anemia, mono-cytosis, eosinophilia, neutropenia or neutrophilia hypoalbuminemia; evidence of vertical transmission	181	[26,29,30,56,96,103,112,126,127,128,129,130,131,132,133,134,135,136]
	Cat	Thrombocytopenia, anemia, or leukopenia/ leukocytosis	14	[135]
*Candidatus Neoehrlichia mikurensis*	Rodent	No symptoms	1/34 (2.9%)	[125]
*Ehrlichia canis*	Cat	No symptoms	2/85 (2.3%)	[19]
	Fox	No symptoms	113/225 (52–56%)	[137,138]
	Dog	Pulmonary hypertension, fever, anemia, tongue ulcer, lymphadenopathy, polyclonal gammopathy, weight loss, anorexia, dermatitis, epistaxis	78	[111,126,132,133,134,136,139,140,141,142]
	Gray wolves	No symptoms	3/6 (50%)	[138]
*E. ovina*	Sheep	No symptoms	1/87 (1.1%)	[10]
*Ehrlichia *spp.**	Cat	lymphadenopathy, pale mucous, stomatitis, sign of respiratory involvement	14/260 (5.3%)	[143]
*Rickettsia conorii*	Dog	Fever, anemia, thrombocytopenia, leukocytosis, hunched posture, abdominal pain, orchitis, splenomegaly, lymphadenopathy, vomiting, diarrhea, hyperglobulinemia, elevated liver enzyme	27	[106,112,133,144]
	Eurasian Otter	Carcass	1/1 (100%)	[145]
*Rickettsia *spp.**	Cat	limp adenomegaly, pale mucous, stomatitis, sign of respiratory involvement	83/260 (31.9%)	[143]

**Table 4 pathogens-10-00181-t004:** *Rickettsiales* identified by molecular methods in clinically ill patients in Italy.

Species	Fever	*Tache* *Noire*	Rash	Clinical Notes	Number of Cases	Reference
*Rickettsia* *conorii*	Yes	Yes	Yes	MSF, Sepsis, respiratory insufficiency and quadriplegia	1	[146]
	Yes	Yes	Yes	MSF, PCR positive on eschar while negative on whole blood	1	[147]
	Yes	Yes (two)	Yes	MSF, Rhabdomyolysis, acute kidney injury and Encephalitis	1	[148]
	Yes	Yes	Yes	MSF, acute kidney injury and herpetic esophagitis	1	[149]
	Yes	Yes	Yes	MSF, Myocarditis-sepsis induced multi organ failure	1	[150]
	Yes	Yes	Yes	MSF	5	[151]
	Yes	Yes	Yes	MSF, HIV patient	1	[152]
	Yes	Yes	Yes	MSF	1	[153]
*R. conorii *subsp.* indica*	Yes	Yes	Yes	MSF	1	[154]
*R. conorii *subsp.* israelensis*	Yes	No	Yes	Petechial rash, severe sepsis and multi organ failure	1	[155]
	Yes	No	Yes	Dysarthria, dysdiadochokinesis, mild neck stiffness, meningism	1	[155]
	Yes	Yes	Yes		1	[156]
*R. massiliae*	Yes	Yes	No	TIBOLA, bite on eyelid, PCR positive on lesion and swab	1	[157]
	Yes	Yes	No	TIBOLA; palpable liver, PCR positive on eschar	1	[158]
	Yes	Yes	Yes	MSF	1	[159]
*R. monacensis*	Yes	Yes	No	MSF, PCR positive on *tache noir* and negative in whole blood	1	[160]
*R. africae*	Yes	Yes	Only ipsilateral limb	Sacral syndrome, traveler from Zimbabwe	1	[161]
*R. slovaca*	No	No	No	asymptomatic	1	[92]
	Yes	Yes	No	TIBOLA, alopecia and painful lymph node	1	[92]
	No	Yes	No	TIBOLA, myalgia, weariness and painful lymph node	1	[92]
	No	Yes	No	TIBOLA, itching	1	[92]
	No	Yes	No	TIBOLA, painful cervical lymph node	1	[92]
	Yes	Yes	Yes	MSF-like	1	[154]
*R.* *aeschlimannii*	Yes	Yes	No	Hepatitis, PCR positive on liver biopsies	1	[162]
*Anaplasma* *phagocytophilum*	Yes	No	Yes	Atypical pneumonia, leukopenia and thrombocytopenia, high liver enzyme	1	[163]
	Yes	No	Yes	Oral erythema, edema of labium, leukopenia and thrombocytopenia	1	[163]
	Yes	No	No	6-month illness-misdiagnosis	1	[164]
	Yes	No	No	Myalgia	1	[165]
	Yes	No	No	Nuchal rigidity and myalgia	1	[165]
	No	No	No	Asymptomatic	1	[165]

## Data Availability

No new data were created or analyzed in this study. Data sharing is not applicable to this article.

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
