# Peer review of "Rickettsiales in Italy"

_pathogens, 2021, doi:10.3390/pathogens10020181_

Round 1

Reviewer 1 Report

Details in attachment.

Author Response

Comments and suggestions:

General:

  1. What is the title of the manuscript? “Rickettsiales in Italy” or “Rickettsiales in Italy. A Systematic Review”. It is different in the manuscript and different in the submission system.

The title of manuscript was changed as “Rickettsiales in Italy. A Systematic Review”

  1. The chapter Introduction needs to be rewritten. Please clearly explain the division within the taxonomic order of Rickettsiales using relevant references, e.g. Szokoli F, Castelli M, Sabaneyeva E, Schrallhammer M, Krenek S, Doak TG, Berendonk TU, Petroni G. Disentangling the Taxonomy of Rickettsiales and Description of Two Novel Symbionts ("Candidatus Bealeia paramacronuclearis" and "Candidatus Fokinia cryptica") Sharing the Cytoplasm of the Ciliate Protist Paramecium biaurelia. Appl Environ Microbiol. 2016 Nov 21;82(24):7236-7247. doi: 10.1128/AEM.02284-16. PMID: 27742680; PMCID: PMC5118934

The introduction was rewritten and the suggested citation was added

  1. Please clearly state the purpose of the work and explain why the authors only focused on the genera Anaplasma, Ehrlichia, Orientia, and Rickettsia (p.2/58-59).…

Done

  1. In the chapter Discussion is: “...We focused our search on the genera of Rickettsiales pathogenic for humans ....” (p.13/102-103) , so why authors described species occurrence only in vectors or animals e.g. Anaplasma platy “... A.platy, found only in ticks, were found both in ticks and fleas .....” (p.13/108-109). This inaccuracy is probably due to the lack of a clearly defined goal of the work

The goal of the work was also focusing attention on Rickettsiales found in bloodsucking arthropods that could potentially attack humans and cause them infections

In the chapter Introduction: “... Our research was therefore principally focused on the genera Anaplasma, Ehrlichia, Orientia, and Rickettsia.” (p.2/58-59). Why is there no data to the genus Orientia in the text? Please explain.

We have added in the result that “never Orientia spp. were found in Italy” to explain the absence of this genera in the review

  1. In Introduction is: “... Rickettsial diseases continue to be the cause of serious health problems in Italy...” (p.1/44), but the authors do not provide data to support this claim (e.g., How many cases of Rickettsiales diseases in humans and domestic animals are reported annually? Do they cause high mortality?). Is it a health problem from a medical or veterinary point of view, or both?…

The text was changed and epidemiological data were added

  1. All the Latin names of the species, which are used in the text, should be written in italics. Correct throughout the text.

Done

  1. Disease names: e.g. Human Monocytic Ehrlichiosis should be written in lower case. Correct throughout the text.

Done

  1. Please provide the full name of the species only when it appears for the first time in the text. Then use the abbreviation of the genus name, e.g. A. platy. Correct throughout the text.

Done

  1. Please provide the full name of the diseases only when it appears for the first time in the text. Then use the abbreviation e.g. Mediterranean spotted fever (MSF). Correct throughout the text.

Done

  1. When the abbreviation “spp.” is used, please put a dot at the end of it. Correct throughout the text.

Done

Title:

p.1/2 Latin name of the species - should be in italics

Done

p.1/2 correct the title: “Rickettsiales in Italy” or “Rickettsiales in Italy. A Systematic Review

We had corrected Rickettsiales with Rickettsiales

Abstract:

p.1/15 "...Ricetett-siales..." - should be: ....Ricetettsiales....

Done

p.1/17 "... Micro-organism..." - should be: ....Microorganism....

Done

Introduction

p1/26-27 reference is needed…

Done

p.1/28-29 “... helmintis...” - should be: helminths

Done

p.1/33 - reference is needed

Done

p.2/47 “... Tibola / Debonel...” – give a full names

we put in brackets the full name of the disease

p.2/49-51  reference is needed?

Materials and Mathods:

p.2/64-65 “... The Preferred Reporting Items for Systematic Reviews and Meta-Analyses (PRISMA) methodology...” - reference is needed

Done

p.2/65-66 “...Rickettsiales identified by PCR ....” – which type of PCR? (conventional, nested, Real-time, all of them) - so better: by molecular methods (and give a details)

pcr quale?

Results

p.2/71 “...these 216 were duplicate....” – What is correctly? In the Figure is: “n= 220” duplications

the data in the figure was correct, we correct in the text, the duplicate was 220

p.2/73-74 “....finally, 167 published from 1997 to 2021studies were included in this review...” What is correctly? In the Figure is: “157 studies were included...” p.2/75-80

data were corrected

p.2/75-80 “...The results of our search could be divided in four sections and are analytically reported in Table 1-4. .....” – the sentences should be rephrased. Data analysis should be carried out in the text and tables help to bring the data together and better visualize it.

The sentence was rephrased

p.3/81 – “... 36 different Rickettsiales species .....” - please give a more details e.g. from which genera (please list them); give a reference to the appropriate Table

Done

p.3/84 “...Rickettsiales were identified in 179 different animal infections..” - please give a more details e.g. symptomatic or asymptomatic infections; give a reference to the appropriate Table

Done

p./87-88 “...Rickettsiales were identified in 29 species of arthropods, most of them were Ixodidae ticks and 4 species of fleas....” - give a reference to the appropriate Table

Done

p.89/90 “....Rickettsiales were found in 15 species of animals with or without symptoms, most of them where A.phagocytophilum and A.platy...” - please give a more details; give a reference to the appropriate Table

Done

p.3/91-98 give a reference to the appropriate Table

Done

Table 1,2: The table is not readable, please add an additional column: Reference. The title column “Host” should be change, because “vegetation” and” free life” are not a host, the same in Table 2. Table

Done

Table 3 “Species of Rickettsiales indetified un wild and domestic animals”. - which means “un”

“un” was “in”, it was an error during digitation. Corrected.

Table 4: please give a number of cases and diagnostic method; change “Authors” for Reference and delete the names of authors, but leave only numbers; “Specie”, “Tache noir”, “Clinical singularity” – unclear, rephrase

Done

Table 2, Table 3, Table 4, Table 5 –explanation of abbreviations (SE,CI, p (value?), OR, Ref, §, #) is necessary ???

All unnecessary abbreviations were removed

Discussion

p.13/116-117 “.. it was found very often in Ixodes ticks. Ixodes is the most diffused tick genera in Italy...” - what species of Ixodes?

we added: “of these the most common was I.ricinus” to be more specific on the species of the most common tick.

p.14/155 “...tache noir..” – please explain what is this?

we added in the results a short definition of the tache noir

p.15/133 -195 ???

p.15/197 reference is needed ?

p.15/205 “...CNS...” – give a full name

Done

p.16/239-241 – “R.conorii subsp israeliensis” or R.conorii subsp. Israeliensis - it should be the same in the text and in tables

We correct all R.conorii subsp.israelinsis with R.conorii subsp. Israeliensis

p.17/284 “...Hippoboscidae Cataerina pallid...” - Crataerina pallida is a species name which is member of Hippoboscidae dipteral

we corrected the species name with H.Crataerina pallida in the text

p.17/302 reference is needed

we added this reference: “Raoult, D.; Berbis, P.; Roux, V.; Xu, W.; Maurin, M. A New Tick-Transmitted Disease Due to Rickettsia Slovaca. Lancet 1997, 350, 112–113, doi:10.1016/S0140-6736(05)61814-4.”

p.17/307-308 “...Ixodes[13,50,53], Rhipicephalus[44] and Hyalomma[84]” – give the species names or insert “spp.”

Done

P17/323 “yaaf” – explain Yaaf

Done

P18/328-329 reference is needed

Done

p.18/345 “ ... H.Cataerina pallid...” – change for correct form

Done

p.18/348-349 reference is needed

Done

Reviewer 2 Report

Overall, this is a thorough and well-written review with some editing needed. The study describes Rickettsiales in Italy based on Pubmed searches conducted prior to December 31, 2020 and summarizes the information obtained.

Overall comments:

Bacterial names need to be in italics throughout the manuscript. This is done inconsistently.  Many places lack italics. 

Expand the introduction slightly to cover the pathogenesis and life cycle of Rickettsiales - how do they infect and a small description of the vectors and hosts and how they are spread. This doesn't need to be long but it will help the reader before diving into the bulk of the date presented

Tache noir,  Rush, TIBOLA should be described when first introduced in either the results to discussion. A simple explanation of the infection would suffice but should be included for the reader

Figure 1 is missing a legend. Please add one.

Figure 1 needs to have the font sized increased slightly and a better quality JPEG needs to be used. The current one has poor resolution when zoomed into.

Specific comments:

Line 47: Include a definition or description of Tibola and Debonel

Lines 85-95 are very choppy and difficult to follow. Authors need to consider revising. They also only really summarize the data from Table 1. Very little effort was used to include any potential summary from Tables 2-4.  Please expand the results section to better cover the spectrum of data presented.

Line 96 : why is arthropods in bold?

Line 96: Table title is not very accurate.  Authors should consider revising. Suggested title:  Table 1. Species of Rickettsiales,  arthropods from which they were isolated and corresponding hosts

Line 97: Reword title of Table 2 to also include the host

Line 142: "l1eukocytosis".  Do the authors mean "leukocytosis"?

Author Response

Bacterial names need to be in italics throughout the manuscript. This is done inconsistently. Many places lack italics.

We had corrected and now the names are in Italics

Expand the introduction slightly to cover the pathogenesis and life cycle of Rickettsiales - how do they infect and a small description of the vectors and hosts and how they are spread. This doesn't need to be long but it will help the reader before diving into the bulk of the date presented

The requested information was added in the introduction and in the discussion

Tache noir, Rush, TIBOLA should be described when first introduced in either the results to discussion. A simple explanation of the infection would suffice but should be included for the reader

Done

Figure 1 is missing a legend. Please add one.

Done

Figure 1 needs to have the font sized increased slightly and a better quality JPEG needs to be used. The current one has poor resolution when zoomed into.

Done

Line 47: Include a definition or description of Tibola and Debonel

Done

Lines 85-95 are very choppy and difficult to follow. Authors need to consider revising. They also only really summarize the data from Table 1. Very little effort was used to include any potential summary from Tables 2-4. Please expand the results section to better cover the spectrum of data presented.

Done

Line 96 : why is arthropods in bold?

It was an error. Corrected

Line 96: Table title is not very accurate. Authors should consider revising. Suggested title: Table 1. Species of Rickettsiales, arthropods from which they were isolated and corresponding hosts

Done

Line 142: "l1eukocytosis". Do the authors mean "leukocytosis"?

Yes, corrected

Round 2

Reviewer 1 Report

The manuscript requires minor revision, especially in the chapters: Results, Conclusion and Tables 3, 4. Details in the attachment. 

Author Response

Review of a manuscript “Rickettsiales in Italy. A Systematic Review”by authors: Cristoforo Guccione, Claudia Colomba, Manlio Tolomeo, Chiara Iaria, Antonio Cascio

 Manuscript ID: pathogens-1078371

Round 2

The comments are detailed below

Comments and suggestions:

Introduction: p.2/52-53 —"....and many other Rickettsia spp. or subspecies have been identified in recent years in humans, vector arthropods and animals.” —reference is needed

We cited this article “Parola, P.; Raoult, D. Ticks and Tickborne Bacterial Diseases in Humans: An Emerging Infectious Threat. Clinical Infectious Diseases 2001, 32, 897–928, doi:10.1086/319347.”

p.2/54-55 — “Other rickettsioses that have been historically documented in Italy are murine typhus and epidemic typhus...” - reference is needed

We cited this articile “Ciceroni, L.; Pinto, A.; Ciarrocchi, S.; Ciervo, A.              . Annals of the New York Academy of Sciences 2006, 1078, 143–149, doi:10.1196/annals.1374.024.”

Materials and Methods:

p.2/71— “Rickettsiales identified by PCR…”— which type of PCR? (conventional, nested, Realtime, all of them) — so better: by molecular methods (and give a details)

To clarify this issue, we added the following sentence “Only studies that provided data about Rickettsiales identified by molecular methods in Italy were included in the review. All molecular methods which reached the species level were considered..”

Results:

p.3/84-89 In order to improve the readability of the results, it is proposed to divide the chapter into subchapters and place appropriate tables under each subchapter. Delete sentences p.3/84-89.

For example:

3.1 Rickettsiales and arthropod vectors

3.2 Rickettsiales identified in animals

3.3 Rickettsiales involved in human disease

We deleted the sentences p.3/84-89, divide the chapter into subchapters as you had suggested reorder the chapter by placing the table after the respective subchapter

p.3/92—" (see Table 1-4)” — delete: see

Done

p.3/94—" (see Table 2)” — delete: see Table 2

Done

“Dermacentor reticularis” — change for Dermacentor reticulatus

Done

Table 3 —please give a number of cases, and in screening give a % of positive samples (prevalence);

We added the column “Number of case(%prevalence)”. When the studies was done only about clinical ill animal we report only the number of case. Indeed, when the cohort was larger or the aim of the study was the screening we report the total number of tested animals and the percentage of prevalence. We clarify this issue also in the text when we describe Table 3. Furthermore, when the screening study was more than one we report the highest and lowest ones.

Table 4 — please give a number of cases or the title of Table 4 should be rewritten, because it is still unclear how many symptomatic cases were identified, please give the method used in confirmation of the pathogens (PCR, serology ?)

We has rewritten the title of the Table 4 as follow “Table 4. Rickettsiales identified with molecular methods in clinically ill patients in Italy.” The column “Number of case” was added.

Discussion p.12/139 —"Ha. Punctata” change for Ha. Punctata —check all the text

Done

 Conclusion Please re-edit the conclusions that strictly refer to the sections analyzed in the text. Please don’t describe what the Authors done in this review but shortly what are the conclusions from results of work.

Rickettsiales found in humans in Italy were: R. aeschlimannii, R. africae, R. massiliae, R. monacensis, R. slovaca, R. conorii, R. conorii subsp. israelensis, R. conorii subsp. indica and A. phagocytophilum. MSF and TIBOLA and HGA were the most frequent clinical mani-festations. E. canis, A. platy and A. phagocytophilum were the most frequently identified Rickettsiales found in dogs and cattle, respectively. Other Rickettsiales identified were: A. bovis, A. ovis, A. marginale, A. centrale, A. platy, E. ovina, Candidatus N. mikurensis, Candidatus R. siciliensis, Candidatus R. barbariae, Candidatus. R. mendelii, R. hoogstraalii, R. limoniae, R. peacockii, R. rhipicephali, R. sp. Strain S, R. sp. strainTwKm01, R. belli, R. tamurae, R. rioja, R. limoniae, R. raoultii R. honei; some of them, even if it has not yet been demonstrated, could in the future be shown to be capable of causing in humans not yet well characterized syndromic pictures. That's why molecular studies for the search for Rickettsiales should be routinely performed in people who have been bitten by bloodsucking arthropods.